# Mitochondrial Lipid Peroxidation Is Responsible for Ferroptosis

**DOI:** 10.3390/cells12040611

**Published:** 2023-02-13

**Authors:** Konstantin G. Lyamzaev, Alisa A. Panteleeva, Ruben A. Simonyan, Armine V. Avetisyan, Boris V. Chernyak

**Affiliations:** 1Belozersky Institute of Physico-Chemical Biology, Lomonosov Moscow State University, 119991 Moscow, Russia; 2The “Russian Clinical Research Center for Gerontology” of the Ministry of Healthcare of the Russian Federation, Pirogov Russian National Research Medical University, 129226 Moscow, Russia

**Keywords:** ferroptosis, mitochondria, complex I, mtROS, mitochondria targeted antioxidants

## Abstract

Ferroptosis induced by erastin (an inhibitor of cystine transport) and butionine sulfoximine (an inhibitor of glutathione biosynthesis) was prevented by the mitochondria-targeted antioxidants SkQ1 and MitoTEMPO. These effects correlate with the prevention of mitochondrial lipid peroxidation, which precedes cell death. Methylene blue, a redox agent that inhibits the production of reactive oxygen species (ROS) in complex I of the mitochondrial electron transport chain, also inhibits ferroptosis and mitochondrial lipid peroxidation. Activation of ROS production in complex I with rotenone in the presence of ferrous iron stimulates lipid peroxidation in isolated mitochondria, while ROS produced by complex III are ineffective. SkQ1 and methylene blue inhibit lipid peroxidation. We suggest that ROS formed in complex I promote mitochondrial lipid peroxidation and ferroptosis.

## 1. Introduction

Ferroptosis is an iron-dependent form of regulated cell death mediated by lipid peroxidation in cell membranes [1,2]. In recent years, ferroptosis has attracted increased interest as several important anti-cancer drugs have been found to cause this form of cell death, while excessive ferroptosis has been associated with various pathologies. In the pioneering work of Stockwell and co-workers [3], where the term ferroptosis was introduced, it was shown that cells lacking mitochondrial DNA (ρ^0^ cells obtained by rigorous selection in the presence of ethidium bromide) are susceptible to ferroptosis in the same way as parental cells. It was concluded that mitochondrial activity is not required for the execution of ferroptosis. However, later, using a much milder protocol based on mitophagy stimulation, it was shown that ferroptosis induced by cystine starvation or by the cystine uptake inhibitor erastin was significantly attenuated in cells with depleted mitochondria [4]. At the same time, mitochondrial depletion had no effect on ferroptosis induced by RSL3, an inhibitor of glutathione peroxidase 4 (GPx4), which detoxifies lipid peroxides. [4]. In the same study, electron transport chain (ETC) inhibitors and the uncoupler of oxidative phosphorylation were shown to also suppress erastin-induced but not RSL3-induced ferroptosis. In contrast to these results, earlier it was reported that erastin effectively induced ferroptosis in cells depleted of mitochondria as a result of mitophagy stimulation [5] (and that an ETC complex I inhibitor promotes lipid peroxidation and ferroptosis [6]). Very recently, mitochondrial DNA depletion by selection in the presence of ethidium bromide has been reported to prevent ferroptosis induced by both erastin and RSL3, presumably due to increased expression of mitochondrial GPx4 [7]. These conflicting results indicate a complex and not fully understood the role of mitochondria in ferroptosis.

Lipid peroxidation is a critical event in ferroptosis [3], but the role of mitochondrial lipid peroxidation remains poorly understood. GPx4 inactivation has been reported to induce lipid peroxidation in some cell membranes but not in mitochondria [8]. In the same study, the mitochondria-targeted antioxidant MitoQ (ubiquinol conjugated with triphenylphosphonium cation) was shown to inhibit ferroptosis, but its non-targeted counterpart, idebenone, was even more effective. MitoQ and the nitroxide-based mitochondria-targeted antioxidant MitoTEMPO were recently reported to prevent mitochondrial lipid peroxidation and ferroptosis induced by both erastin and RSL3, while decylubiquinion was found to be ineffective [7]. MitoTEMPO has also been reported to prevent doxorubicin-induced cardiac ferroptosis in mice [9]. Another nitroxide targeted to mitochondria by conjugation with hemigramicidin S, XJB-5-131, also inhibits ferroptosis much more effectively than its non-targeted analog JP4-039 [10]. Recently, dihydroorotate dehydrogenase (DHODH) has been shown to protect against ferroptosis induced by GPx4 inactivation by inhibiting mitochondrial lipid peroxidation [10,11]. DHODH is located on the outer surface of the inner mitochondrial membrane and catalyzes the oxidation of dihydroorotate associated with the reduction of coenzyme Q (CoQ) to CoQH2, which detoxifies lipid peroxyl radicals in mitochondria. More recently, a similar protective antiferroptotic mechanism has been described [12] based on the reduction of CoQ by mitochondrial glycerol-3-phosphate dehydrogenase 2 (GPD2). These studies clearly point to the role of mitochondrial lipid peroxidation in ferroptosis.

In the present study, we demonstrate that the mitochondria-targeted antioxidants SkQ1(10-(6′-plastoquinonyl) decyltriphenylphosphonium bromide) and MitoTEMPO inhibit ferroptosis induced by either erastin or glutathione depletion. Using a novel lipid peroxidation-sensitive mitochondrial-targeted fluorescent probe (MitoCLox), we show that SkQ1 prevents the mitochondrial lipid peroxidation that precedes cell death in these models. Using an in vitro model of ferroptosis, it is shown that reactive oxygen species (ROS) produced by ETC complex I induces iron-dependent lipid peroxidation in isolated heart mitochondria, while the ROS produced by complex III is ineffective. SkQ1 effectively prevents mitochondrial lipid peroxidation in this model.

## 2. Methods

### 2.1. Chemicals

MitoCLox was synthesized from 4-difluoro-5-(4-phenyl-1,3-butadienyl)-4-bora-3a,4a-diaza-s-indacene-3-propionic acid succinimidyl ester (C11-BODIPY581/591 SE) (Invitrogen Life Technologies, Waltham, MA, USA) and {5-[(4-aminobutyl)amino]-5-oxopentyl}(triphenyl)phosphonium bromide as described in [13]. C11-BODIPY581/591 is a ratiometric fluorophore that specifically reacts with lipid peroxy radicals to change the fluorescence emission maxima from red to green. It was shown that after conjugation of this fluorophore with the penetrating triphenylphosphonium cation, it selectively accumulates in the mitochondria of living cells and registers mitochondrial lipid peroxidation [13,14]. SkQ1 and dodecyltriphenylphosphonium bromide (C_12_TPP) were kindly provided by the Institute of Mitoengineering, Lomonosov Moscow State University. Other reagents, except for those indicated, were from Sigma-Aldrich (Saint Louis, MO, USA).

### 2.2. Cell Cultures

Human fetal lung fibroblasts MRC5 transformed with SV-40 (MRC5 SV2, EcACC Cat. No. 84100401) and primary skin fibroblasts from Common Use Center “Biobank” (Research Centre for Medical Genetics, Moscow, Russia) were cultured in DMEM (Dulbecco’s modified Eagle’s) medium (Gibco; Thermo Fisher Scientific, Inc., Waltham, MA, USA) supplemented with 2 mM glutamine and 10% fetal bovine serum (FBS) (HyClone, Logan, UT 84321 USA) and 100 U/mL streptomycin and 100 U/mL penicillin (all from Gibco, USA). Cell viability was measured using the CellTiterBlue^®^ reagent (Promega, USA) according to the manufacturer’s protocol with Fluoroskan Ascent FL Microplate Reader (Thermo Labsystems, Waltham, MA, USA).

### 2.3. Microscopy

Human fetal lung fibroblasts MRC5-SV40 were plated in 35 mm glass bottom (SPL) dishes for confocal microscopy at 150,000 cells. Cells were incubated with erastin (10 mM) for 17 h. After incubation, cells were stained with 200 nM MitoCLox for 2 h and analyzed using an Axiovert 200 microscope (Carl Zeiss, Germany)

### 2.4. Flow Cytometry

MRC5-SV40 or primary human fibroblasts cell lines were incubated with erastin (10 mM) or BSO (1 mM) for 17 h, and then cells were stained with 100 nM MitoCLox (1 h) or 1.8 μM DCFH2 DA (30 min). After this, cells were stripped with trypsin/versene, and the medium containing cells was centrifuged in 1.5 mL tubes (1000 r min^−1^, 3 min) at 4 °C. The centrifuged cells were redispersed in PBS (0.05 mL). Each sample was measured until 2000 events had been collected. Flow cytometry analyses were performed using an Amnis FlowSight Imaging Flow Cytometer (Luminex Corporation, Seattle, WA, USA) with excitation at 488 nm and the detection channels 505–560 (Ch2) and 595–642 (Ch4). For ratiometric analysis, the Amnis IDEAS^®^ 6.2 (Luminex, Seattle, WA, USA) image analysis software was used.

### 2.5. Isolated Mitochondria

Mitochondria were isolated from rat hearts. The isolation medium contained 250 mM sucrose, 5 mM Mops KOH, pH 7.4, 1 mM EGTA, and BSA (0.5 mg/mL). The heart was minced with scissors in a cooled isolation medium in a ratio of 10 mL/g of cardiac tissue. The suspension was supplemented with Nagarse (type XXIV; Sigma-Aldrich) and homogenized in a glass Potter homogenizer for 1–2 min. The homogenate was diluted with an isolation medium to the ratio of 20 mL/g of original tissue and centrifuged for 10 min at 600× *g*. The supernatant was collected and centrifuged for 10 min at 12,000× *g*. The precipitate was resuspended in a minimum volume, homogenized, diluted in 20 mL of isolation medium, and centrifuged for 10 min at 12,000× *g*. The pellet was resuspended and stored on ice.

The lipid peroxidation measured in the medium contained 250 mM sucrose, 5 mM Mops KOH, pH 7.4, 10 μM FeSO_4,_ and 50 nM MitoClox at 25 °C.

H_2_O_2_ production was measured fluorimetrically using 10 μM Amplex Red and 4 U/mL horseradish peroxidase at 25 °C, excitation/emission 57/585 nm.

Additions: 5 mM glutamate/5 mM malate (G/M), 10 μM rotenone, 5 mM succinate, 2 μM antimycin A. Concentration of mitochondrial protein—0.4 mg/mL as determined by Bradford method.

### 2.6. Statistics

At least three repeats for each measurement were performed. Results are presented as the mean of a minimum of 3 independent replicates with standard deviation (SD). Comparisons were analyzed by one-way ANOVA. The significance was analyzed with Prism 7.0 software (GraphPad Software, LLC, California, USA); a value of *p* < 0.001 was considered to be statistically significant.

## 3. Results

### 3.1. Mitochondria-Targeted Antioxidant SkQ1 Inhibits Ferroptosis Induced with Erastin

Induction of ferroptosis with erastin was studied on MRC5 human lung fibroblasts transformed with the SV40 virus (MRC5-SV40). Erastin-induced necrotic death of these cells, which was prevented by the antioxidant α-tocopherol (α-toc), the intracellular iron chelator deferoxamine (DFO), and the specific ferroptosis inhibitor ferrostatin-1, indicating ferroptosis as the main mechanism of cell death (Figure 1a). The addition of SkQ1 simultaneously with erastin strongly inhibited ferroptosis, while its analog lacking the antioxidant quinone moiety (C_12_TPP) was ineffective, indicating an important role of mitochondrial ROS (mtROS) in erastin-induced ferroptosis. (Figure 1b). Another mitochondria-targeted antioxidant, MitoTEMPO, also prevented cell death (Figure 1a). As the SkQ1 concentration increased above 200 nM, the protective effect decreased (Figure 1b). A similar dose–effect relationship has been observed previously for protection against H_2_O_2_-induced apoptosis with SkQ1 and MitoQ [15]. This correlates with increased endogenous ROS production and is explained by the prooxidant action of quinone-based compounds.

The redox cycling agent methylene blue (MB) also targets mitochondria due to its positive charge [16]. MB is reduced by complex I and oxidized by cytochrome c, thus bypassing the proximal part of the ETC [17]. The reduced MB is also directly oxidized by oxygen to form hydrogen peroxide. As shown in Figure 1a, MB effectively suppressed erastin-induced ferroptosis. These data are in complete agreement with a recent study that reported protection against erastin-induced cell death by MB and its analogs in primary fibroblasts obtained from a patient with Friedreich’s ataxia [18]. The antiferroptotic effect of MB can be explained by the prevention of mtROS production by complex I. Data confirming this possibility were obtained in experiments with isolated mitochondria (see below). At the same time, other mechanisms of MB action cannot be excluded. For example, MB-induced production of hydrogen peroxide can stimulate the transcription factor Nrf2 (nuclear factor erythroid 2-related factor 2), followed by the expression of antioxidant enzymes [19].

Erastin-induced ferroptosis of MRC5-SV40 cells was not affected by inhibition of complex I by piericidin A or rotenone, inhibition of complex III by antimycin A, or uncoupling of oxidative phosphorylation by carbonyl cyanide 4-(trifluoromethoxy)phenylhydrazone (FCCP) (Figure 1a). These data contradict early results [4], probably due to differences in glutathione metabolism associated with mitochondrial functions in different cells (see below).

### 3.2. Ferroptosis Induced by Glutathione Depletion in Fibroblasts Derived from Patient with Leber’s Hereditary Optic Neuropathy (LHON) Depends on mtROS Production

Reduced glutathione (GSH) is an important component of cellular defense against ferroptosis, primarily as a substrate for GPx4 [20]. Moreover, the mitochondrial GSH pool may be especially important for protection against ferroptosis [21]. We used the gamma-glutamylcysteine synthetase inhibitor butionine sulfoximine (BSO) to stimulate glutathione depletion but did not observe significant toxicity in MRC5-SV40 up to millimolar concentrations. We tested several primary fibroblasts derived from the skin of healthy donors but again only observed low BSO toxicity.

Next, we analyzed fibroblasts obtained from a patient with Leber’s hereditary optic neuropathy (LHON) and found significant BSO toxicity. BSO induced necrotic death in these cells, which was prevented by DFO and ferrostatin-1, confirming ferroptosis (Figure 2a). We showed that the pan-caspase inhibitor zVADfmk did not affect cell viability, indicating that caspase-dependent apoptosis was not responsible for BSO-induced cell death. SkQ1, but not C_12_TPP, strongly inhibited ferroptosis (Figure 2b), and the dose-response relationship was similar to that observed in erastin-induced ferroptosis. MB also prevented cell death (Figure 2a), indicating an important role for complex I-produced mtROS in ferroptosis induced by GSH depletion. Accordingly, fibroblasts from LHON patients appeared to be significantly more sensitive to erastin-induced ferroptosis than healthy fibroblasts and MRC5-SV40 fibroblasts. LHON is the most common hereditary mitochondrial disease that primarily results from point mutations in the mitochondrial genes of the ND1, ND4, and ND6 complex I subunits. The patient donating cells in the study carries the mutation 13513G>A [22]. It was previously shown that these cells have an increased level of endogenous oxidative stress [22], which is probably the reason for the high sensitivity to BSO. This finding is consistent with earlier studies showing that BSO induces significantly higher cytotoxicity in fibroblasts from patients with other mitochondrial diseases than in healthy human fibroblasts [23,24]. Hypersensitivity to BSO has been described in fibroblasts from patients with Leigh’s syndrome caused by a mutation in SURF1 (a gene encoding a mitochondrial chaperone involved in cytochrome oxidase assembly) [24] or two mutations in the ND3 and NDUFA1 subunits of complex I [23], as well as in fibroblasts of patients with MELAS (myopathy, encephalopathy, lactic acidosis, and stroke-like episodes) caused by mutations in two mitochondrial tRNAs [23].

### 3.3. Peroxidation of Mitochondrial Lipids Is Critical for Ferroptotic Cell Death

The level of lipid peroxidation in mitochondria during ferroptosis was analyzed using a new mitochondria-targeted fluorescent ratiometric probe MitoCLox [13]. This probe consists of the C11-BODIPY581/591 fluorophore, which reacts specifically with lipid peroxy radicals, changing the fluorescence emission maxima from red to green. The fluorophore is conjugated with the penetrating triphenylphosphonium cation using a linker containing two peptide bonds. A similar probe with a different linker structure was described earlier under the name MitoPerOX [25] and was recently used to demonstrate mitochondrial lipid peroxidation in ferroptosis [7]. The specific oxidation of MitoCLox by lipid radicals, its selective accumulation in mitochondria of various cells, and the response to mitochondrial lipid peroxidation induced by oxidative stress is shown [14].

Since erastin induced massive death of MRC5-SV40 cells at 24 h (Figure 1a), we have measured MitoCLox oxidation at earlier time points. Significant oxidation of MitoCLox in some fractions of viable cells was detected at 17 h (Figure 3). Mitochondrial localization of the oxidized probe was confirmed by fluorescence microscopy (Figure 3a). Quantitative analysis using flow cytometry showed a strong heterogeneity in the level of oxidation in the cell population (Figure 3b). A similar pattern of MitoCLox oxidation was observed in BSO-treated LHON fibroblasts (Figure 2c). SkQ1, MB, and DFO but not C_12_TPP, preventing mitochondrial lipid peroxidation in both models (Figure 2d and Figure 3c). It is important to note that the accumulation of hydrogen peroxide was detected by DCFH2DA much earlier (4 h) than MitoCLox oxidation and did not decrease under the action of SkQ1, MB, and DFO in cells treated with erastin (Figure 3d,e). These data indicate that mitochondrial lipid peroxidation is critical for ferroptosis but cannot answer the question of whether mtROS are the primary inducers of lipid peroxidation responsible for initiating ferroptosis (see discussion below).

### 3.4. Lipid Peroxidation in Isolated Mitochondria Is Catalyzed by Complex I but Not Complex III

Lipid peroxidation in isolated rat heart mitochondria was analyzed using MitoCLox. Probe oxidation associated with ETC activity was observed only in the presence of ferrous ion (Fe^2+^), which presumably indicates the involvement of the Fenton reaction. Incubation of mitochondria with Fe^2+^ and NAD-dependent substrates (glutamate and malate) led to slow changes in the fluorescence spectra of MitoCLox, demonstrating oxidation of the probe (Figure 4a). The addition of the complex I inhibitor rotenone stimulated oxidation (Figure 4b). Thus, the production of ROS on flavin mononucleotide (FMN), which is over-reduced in the presence of rotenone, is responsible for lipid peroxidation [26]. Quite unexpectedly, ROS produced by complex III in the presence of the complex III inhibitor antimycin A did not induce lipid peroxidation (Figure 4c), although the rate of hydrogen peroxide production, in this case, was even higher than in the presence of glutamate, malate, and rotenone (Figure 4e).

SkQ1 dose-dependently prevented lipid peroxidation induced by complex-I-dependent ROS production (Figure 4d). Effective concentrations of SkQ1 were higher than in experiments with cells, presumably due to a higher concentration of isolated mitochondria. MB also prevented lipid peroxidation, consistent with its ability to bypass complex I (Figure 4d). SkQ1 did not inhibit hydrogen peroxide production by complex I in the presence of Fe^2+^ and rotenone (Figure 4e), indicating that its protection against ferroptosis is based on the prevention of mitochondrial lipid peroxidation after ROS production. This conclusion is consistent with the results of measuring hydrogen peroxide in cells treated with erastin (Figure 3d,e).

## 4. Discussion

As shown in Figure 1 and Figure 2, the mitochondria-targeted antioxidants SkQ1 and MitoTEMPO prevent ferroptosis induced by erastin or BSO. These data indicate that some oxidative events initiated in mitochondria are critical for ferroptosis. The nature of these events is not clear. As shown in Figure 2 and Figure 3, mitochondrial lipid peroxidation precedes ferroptotic cell death and is prevented by SkQ1. Thus, we hypothesize that lipid peroxidation is at least one of these critical mitochondrial events.

SkQ1 is positively charged and highly membranophilic; therefore, it is located exclusively in the inner mitochondrial membrane, accumulating there due to the high membrane potential (negative from the side of the matrix) [15]. SkQ1 accumulates in the mitochondria of living cells within 30–40 min [15] and, almost immediately after that, blocks oxidative damage to mitochondria and fragmentation of elongated mitochondria induced by exogenous hydrogen peroxide [15,27]. SkQ1 also prevents ROS accumulation in the cytosol and apoptosis induced by hydrogen peroxide, but these effects require preincubation for 24 h or more (depending on the cell type) [15,28]. The same or even longer (up to 7 days) pre-treatment with MitoQ is required to produce a beneficial effect on fibroblast growth [29]. The authors suggested that this period is necessary for the adaptation of cellular signaling pathways to the antioxidant action of MitoQ but did not discuss why this adaptation is not needed for non-targeted antioxidants. Another possibility may be related to the prooxidant effect of MitoQ and SkQ1, which is inherent in quinone-based antioxidants and can induce the expression of antioxidant enzymes. This assumption was ruled out (at least for SkQ1) since pre-treatment with another antioxidant, Trolox (with its subsequent removal), did not abolish the protective effect of SkQ1 [28]. We hypothesized that SkQ1 slowly accumulates in a small subpopulation of mitochondria, which have a reduced membrane potential and produce the main part of ROS under conditions of oxidative stress, determining cell fate. Such a subpopulation was visualized in fibroblasts [28] and endothelial cells [14] using a potential-sensitive dye, and it was shown using MitoCLox that lipids in these mitochondria are more oxidized than in the surrounding mitochondrial population of the same cell. In support of this hypothesis, we showed that necrotic cell death induced by photodynamic damage to most of the mitochondrial population (using Mitotracker Red as a photosensitizer) was prevented by SkQ1 already after a short (1 h) preincubation [15]. The protection against ferroptosis described here also requires only a short preincubation with SkQ1, indicating that the oxidative events responsible for cell death extend to all (or almost all) mitochondria in the cell. In accordance with this, the analysis of mitochondrial lipid peroxidation induced by erastin (Figure 3) showed that lipids are oxidized in almost all mitochondria of affected cells.

The mitochondrial-targeted antioxidants can protect mitochondrial lipids from oxidation, regardless of the ROS source, so a possible role for mitochondrial ROS in ferroptosis remains hypothetical. Some support for this hypothesis is provided by experiments in which protection against ferroptosis and against mitochondrial lipid peroxidation with methylene blue was observed (Figure 1, Figure 2 and Figure 3). This compound prevents the formation of ROS by complex I by directing the electron flow bypassing the ROS-producing sites of the enzyme [17,30]. Moreover, the nonenzymatic oxidation of MB with oxygen produces significantly more hydrogen peroxide than with complex I [30]. Thus, it is likely that ROS produced by complex I are specifically effective in inducing lipid peroxidation. To test this possibility, we analyzed lipid peroxidation in isolated mitochondria using MitoCLox (Figure 4). It has been shown that lipid peroxidation induced by substrates of complex I is strictly dependent on Fe^2+^, which indicates the involvement of the Fenton reaction, as was shown for ferroptosis [2,3]. Rotenone stimulated lipid peroxidation; therefore, ROS responsible for lipid peroxidation are produced at the FMN-binding site of complex I [26].

As shown in Figure 4, ROS production by complex III (in the presence of antimycin A) did not induce lipid peroxidation. At the same time, the rate of hydrogen peroxide formation under these conditions was the same as in the presence of glutamate, malate, and rotenone. Probably, this difference is explained by the production of ROS in different compartments by the two complexes. If complex I produces ROS in the mitochondrial matrix, then complex III produces most of the ROS on the outer surface of the inner mitochondrial membrane [31,32]. These data suggest that ROS produced by complex III are unlikely to be responsible for lipid peroxidation and cell death during ferroptosis, contrary to the conclusion made by Homma et al. In this study [33], S3QEL, a small molecule that inhibits superoxide production by complex III without inhibition of respiration [34], was shown to prevent cysteine starvation-induced ferroptosis. The effective concentration in this study was 50 µM, in contrast with 1–3 µM, which was found to be effective in another cell model [34]. At high concentrations, S3QEL significantly stimulated GPx4 expression [33], and this effect was probably responsible for the inhibition of ferroptosis, independent of the inhibition of ROS production.

We did not observe the effect of the complex I inhibitors rotenone and piericidin A, the complex III inhibitor antimycin A, and the uncoupler FCCP on cell death during ferroptosis (Figure 1a). These results are inconsistent with both data showing stimulation of ferroptosis by a complex I inhibitor [6] and opposite results showing inhibition of ferroptosis by respiratory inhibitors [4]. Probably, in our models, the counter effects of complex I inhibitors counterbalanced each other. On the other hand, the lack of antiferroptotic effect of mitochondrial inhibitors possibly reflects the difference in the rate of glutathione metabolism between the human fibroblasts used here and the cells (HT1080 fibrosarcoma and mouse embryonic fibroblasts) studied by [4]. The authors explained the action of mitochondrial inhibitors by inhibition of glutaminolysis and the Krebs cycle, which are necessary for ferroptosis since they are involved in the depletion of glutathione reserves during cysteine starvation [4]. Our data showed that MRC5-SV40 fibroblasts and normal subcutaneous human fibroblasts are resistant to GSH depletion by BSO, so it is likely that glutathione metabolism in these cells is slowed down and the critical pool of GSH is insensitive to inhibition of mitochondrial functions. Conversely, in LHON fibroblasts, GSH depletion by BSO is so severe that even inhibition of glutaminolysis cannot induce GSH recovery. These possibilities may be tested in future studies.

Ferroptosis may be associated with various pathologies, including ischemic or toxic lesions of the heart, kidneys, and brain [2]. Autoimmune diseases and pathologies associated with excessive inflammation also probably depend on ferroptosis in immune cells [35]. It has recently been shown that the pathogenesis of some ophthalmic diseases, including dry eye syndrome, may depend on ferroptosis [36]. Iron chelation therapy has long been known and has been suggested to be effective in at least some of the pathologies indicated [37,38]. More specific inhibitors of ferroptosis have not yet reached clinical trials.

Mitochondria-targeted antioxidants have been proposed as a promising therapy for almost the same range of pathologies listed above [39,40]. SkQ1 has been successfully used in preclinical studies for the treatment of cardiovascular and renal diseases [41,42] and also demonstrated anti-inflammatory activity in acute bacterial infection [43] and in the systemic inflammatory response syndrome (SIRS) model [44]. The high efficiency of eye drops containing SkQ1 has been demonstrated not only in various models of eye diseases in animals [45] but also in a clinical study of dry eye syndrome [23]. The coincidence of these two lists of pathologies suggests that inhibition of ferroptosis by SkQ1 may contribute to the therapeutic efficacy of this compound.

## 5. Conclusions

In conclusion, we have shown that mitochondrial lipid peroxidation precedes cell death induced by erastin and BSO. Prevention of mitochondrial lipid peroxidation by mitochondria-targeted antioxidant SkQ1 or by methylene blue that bypasses electron flow through complex I correlate with prevention of ferroptosis. These data indicate that mitochondrial lipid peroxidation mediated by complex I-dependent ROS production is critical for ferroptosis induced by cysteine or GSH depletion. In support of this suggestion, we have shown that ROS produced by complex I but not by complex III can stimulate lipid peroxidation in isolated mitochondria. Mitochondria-targeted antioxidants have proven to be very effective inhibitors of ferroptosis. Due to their low toxicity, these compounds can become an important component in the treatment of pathologies associated with ferroptosis.

## Figures and Tables

**Figure 1 cells-12-00611-f001:**
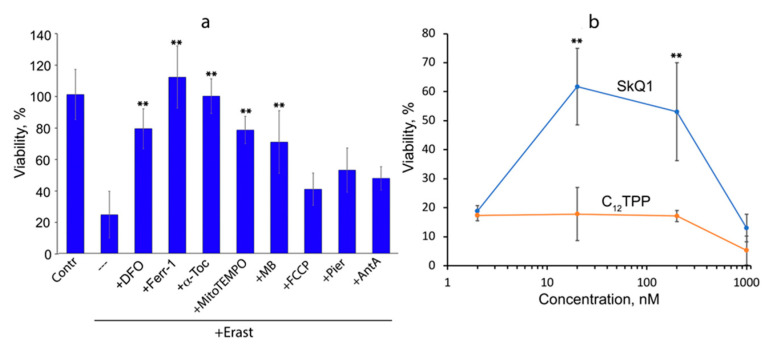
Erastin-induced ferroptosis of MRC5-SV40 cells is inhibited by SkQ1, MitoTEMPO, and methylene blue. MRC5-SV40 cells were treated with erastin (Erast, 10 μM) for 24 h. viability was analyzed using CellTiterBlue^®^ test. (**a**) Deferoxamine (DFO, 100 μM), α-tocopherol (α-toc, 10 μM), ferrostatn-1 (Ferr-1, 25 μM), FCCP (5 μM), piericidine (Pier, 1 μM), antimycin A (AntA, 2 μM), MitoTEMPO (10 μM), and methylene blue (MB, 250 nM) were added together with erastin. (**b**) SkQ1 (10-(6′-plastoquinonyl) decyltriphenylphosphonium bromide) and C_12_TPP (dodecyltriphenylphosphonium bromide) were added together with erastin. **- *p* < 0.001.

**Figure 2 cells-12-00611-f002:**
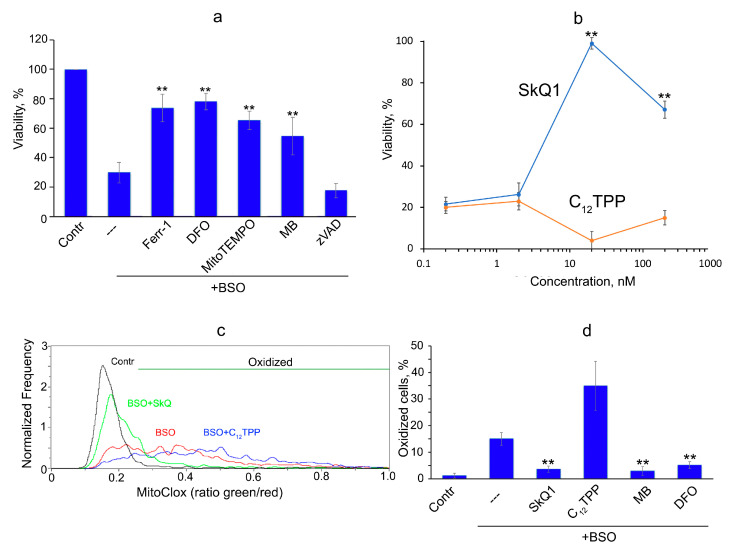
SkQ1 and methylene blue inhibit ferroptosis induced by butionine sulfoximine in fibroblasts from patient with Leber’s hereditary optic neuropathy (LHON). Fibroblasts were treated with butionine sulfoximine (BSO, 1 mM) for 24 h. Viability was analyzed using CellTiterBlue^®^ test. Peroxidation of mitochondrial lipids in fibroblasts was measured with MitoCLox as described in Methods. (**a**) DFO, Fer-1, MitoTEMPO, and MB (concentrations as in Figure 1a), as well as zVADfmk (10 μM), were added together with BSO. (**b**) SkQ1 and C_12_TPP were added together with BSO. (**c**,**d**) Results of MitoCLox oxidation analysis using flow cytometry. Green and red fluorescence were measured simultaneously, and an increase in the green/red ratio indicates oxidation of MitoCLox. Cell fractions with oxidized MitoCLox calculated as shown in (**c**) were measured in three independent experiments (**d**). BSO (1 mM), DFO, MB, SkQ1 (100 nM), and C_12_TPP (100 nM) were added for 17 h. **- *p* < 0.001.

**Figure 3 cells-12-00611-f003:**
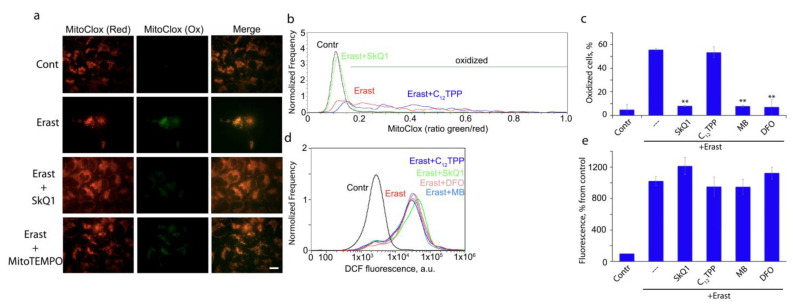
Peroxidation of mitochondrial lipids induced by erastin in MRC5-SV40 cells. Cells were treated with erastin (Erast, 10 μM) for 17 h. Peroxidation of mitochondrial lipids was measured with MitoCLox as described in Methods. (**a**) Fluorescent microscopy of the cells stained with MitoCLox in two channels (green and red). SkQ1 (100 nM) and MitoTEMPO (10 μM) were added together with erastin. Bar, 15 μm. (**b**,**c**) Results of MitoCLox oxidation analysis using flow cytometry. Green and red fluorescence were measured simultaneously, and an increase in the green/red ratio indicates oxidation of MitoCLox. Cell fractions with oxidized MitoCLox calculated as shown in (**b**) were measured in three independent experiments (**c**). MB (250 nM), DFO (100 µM), SkQ1 (100 nM), and C_12_TPP (100 nM) were added together with erastin for 17 h. (**d**,**e**) Results of DCFH2DA oxidation analysis using flow cytometry. SkQ1, C_12_TPP, DFO, and MB were added as in (**b**,**c**) together with erastin for 4 h. Median fluorescence of DCF was measured in three independent experiments. The fluorescence of nontreated cells (Contr) was taken as 100% (**e**). **- *p* < 0.001.

**Figure 4 cells-12-00611-f004:**
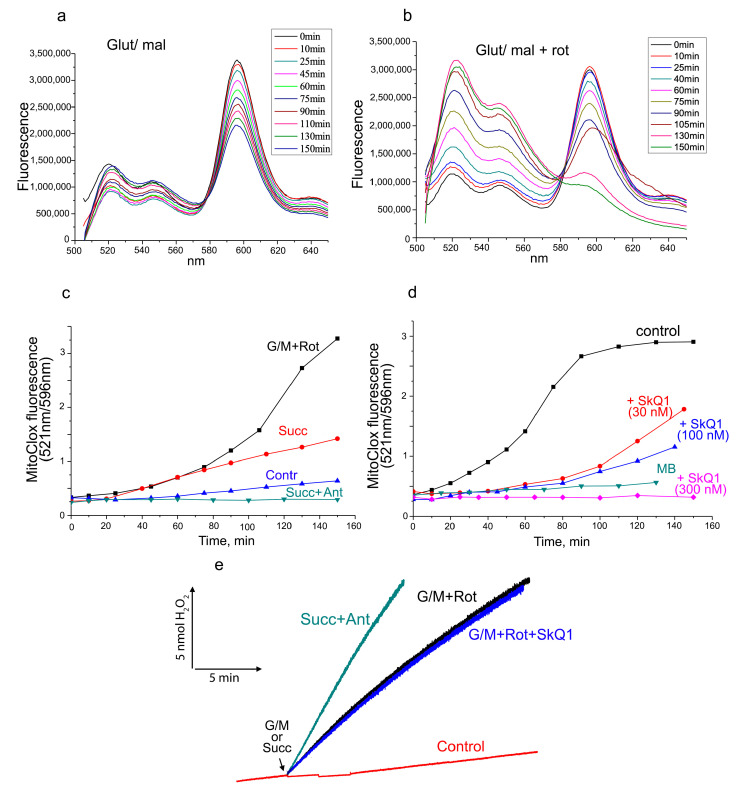
Peroxidation of lipids in isolated rat heart mitochondria in the presence of Fe^2+^. Mitochondria were incubated with MitoCLox (50 nM), and fluorescent spectra were analyzed after subtraction of light scattering as described in Methods. Kinetics of MitoCLox oxidation calculated as a ratio of fluorescence at 521 nm and 596 nm. (**a**,**b**) Changes in fluorescent spectra of MitoCLox with time initiated by glutamate (5 mM) and malate (5 mM) in the absence (**a**) or in the presence (**b**) of rotenone. (**c**) Kinetics of MitoCLox oxidation induced by glutamate, malate, and rotenone (G/M + rot) or by succinate (succ, 5 mM) in the absence or in the presence of antimycine A (Ant). No substrates—Contr. (**d**) The effects of SkQ1 (30 nM, 100 nM, 300 nM) and MB (1 μM) on MitoCLox oxidation induced by glutamate, malate, and rotenone. (**e**) Hydrogen peroxide production was measured fluorometrically using Amplex Red and horseradish peroxidase. The reaction was initiated by glutamate and malate in the presence of rotenone or succinate in the presence of antimycin. SkQ1 (300 nM) was added 5 min before glutamate and malate. No substrates—control.

## Data Availability

The datasets and raw data used and/or analyzed during the current study are available from the corresponding author on reasonable request.

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
