# Peer review of "Mitochondrial Lipid Peroxidation Is Responsible for Ferroptosis"

_cells, 2023, doi:10.3390/cells12040611_

Round 1
Reviewer 1 Report
The paper entitled Mitochondrial lipid peroxidation is responsible for ferroptosis presents some degree of novelty and before being accepted for publication in the journal some issues should be addressed.
In the introduction
The general aim of the work is well-exposed and the references are appropriate. However, some issues in the English language should be accessed some sentences are highlighted where the authors should improve the English language to improve clarity.
“In the pioneering work by Stockwell and co-workers [3], where the term ferroptosis was introduced, it was shown that cells lacking mitochondrial DNA (ρ0 cells obtained by severe selection in the presence of ethidium bromide) are also susceptible to ferroptosis, as well as parental cells, and it was concluded that mitochondrial activity is not required for execution of ferroptosis.”
“At the same time, ferroptosis induced by RSL3, an inhibitor of glutathione peroxidase 4 (GPx4), a peroxidase that detoxifies lipid peroxide and protects against ferroptosis, was not affected by mitochondrial depletion [4].”
Results
Section 2.3
Fibroblasts from LHON patients were used in the experiments conversely to MRC5-SV40 fibroblasts.
The authors add the following sentence “It was previously shown that these cells have an increased level of endogenous oxidative stress [21], which is probably the reason for the high sensitivity to BSO.” – Therefore, when using BSO, an agent that simulates glutathione depletion, and potentially make the cells more prone to oxidative stress damage this results could not be artificially correlated with ferroptosis and instead be correlated with oxidative stress damage and potentially apoptosis?
The authors also write “SkQ1, but not C12TPP, strongly inhibited ferroptosis (Fig. 2b), and the dose-response relationship was similar to that observed in erastin-induced ferroptosis.” – It will be useful also to have since these are different cells than the ones used in the experiments described in section 2.2 have a similar experiment with the cells used in this section.
Section 2.4
The sentence “Quite unexpectedly, ROS produced by complex III in the presence of the complex III inhibitor antimycin A did not induce lipid peroxidation (Fig. 4c), although the rate of hydrogen peroxide production, in this case, was almost the same as in the presence of glutamate, malate, and rotenone.” Some information seems to be missing or not correctly exposed since the observation does not match what is described in the figure.
“SkQ1 dose-dependently prevented lipid peroxidation induced by complex I–dependent ROS production (Fig.4d). Effective concentrations of SkQ1 were higher than in 210 experiments with cells presumably due to a higher concentration of isolated mitochondria.” – In this case should the reverse be expected since as described by the authors in these paper concentrations of SkQ1 at 100 nM or higher have a prooxidant effect due to the plastoquinone nucleus of the compound? In this section, however, the reverse is observed. The authors should better understand these observations since the experiments described in sections 2.1 and 2.2 may suggest the reverse effect.
Other minor modifications/considerations should also be addressed at a later stage after the initial observations are addressed.
Author Response
- The highlighted sentences in the introduction are corrected.
2.”…when using BSO, an agent that simulates glutathione depletion, and potentially make the cells more prone to oxidative stress damage this results could not be artificially correlated with ferroptosis and instead be correlated with oxidative stress damage and potentially apoptosis?”
We analysed the possible role of apoptosis in BSO-induced death of fibroblast from LHON patients. First, we have shown that the pan-caspase inhibitor zVADfmk does not affect cell viability. These data indicate that caspase-dependent apoptosis was not responsible for BSO-induced cell death. These data are included in the corrected version of Fig. 2A. In addition, we analysed the changes in chromatin structure that usually accompany apoptosis. No changes were observed in chromatin stained with Hoechst 33342 (Figure 1 in the attachment ). Thus, apoptosis was not responsible for BSO-induced fibroblast death in LHON patients, and this finding was added to the revised version
- ”It will be useful also to have since these are different cells than the ones used in the experiments described in section 2.2 have a similar experiment with the cells used in this section”.
We have studied the sensitivity of fibroblast from LHON patients to erastin-induced ferroptosis. These cells were shown to be significantly more sensitive to erastin-induced cell death than MRC5-SV40 fibroblasts (Figure 2A in the attachment ). Erastin induced ferroptosis in fibroblasts from LHON patients as necrotic cell death was blocked by ferrostatin-1. SkQ1 also prevents erastin-induced cell death in these cells (Figure 2B in the attachment ). We did not study mitochondrial lipid peroxidation in this model, so these data were not included in the revised version.
- “Some information seems to be missing or not correctly exposed since the observation does not match what is described in the figure”.
On Fig. 4 we have added data on the induction of hydrogen peroxide formation by succinate in the presence of antimycin. Corresponding changes have been made to the text of the revised version.
- “…SkQ1 at 100 nM or higher have a prooxidant effect due to the plastoquinone nucleus of the compound? In this section (2.4), however, the reverse is observed. The authors should better understand these observations since the experiments described in sections 2.1 and 2.2 may suggest the reverse effect”.
SkQ1 is a very lipophilic compound, so its effective concentration is highly dependent on the concentration of phospholipid membranes. In experiments with isolated mitochondria, the concentration of membranes was significantly higher than in experiments with cell cultures, and this is one of the reasons why higher doses of SkQ1 are required for the manifestation of the antioxidant effect. Another possible reason is related to the additional accumulation of SkQ1 (and other cationic compounds) in the cytosol due to the membrane potential (about 60 mV negative inside) on the plasma membrane. At the concentrations of SkQ1 used in experiments with isolated mitochondria, only an antioxidant effect (protection against lipid peroxidation) was observed.

Reviewer 2 Report
This study looked into how ROS formed in complex 1 cause ferroptosis and mitochondrial lipid peroxidation. It is exciting that this research may demonstrate the significance of antioxidants that target the mitochondria in treating illnesses connected to ferroptosis. The discussion of the results is great, and the figures are logical. However, a variety of issues need to be fixed and/or improved before being accepted for publishing.
1. There are more recent articles about BSO, please either update this reference or explain why the authors' findings in result 2.2 are incongruent with reference 20, which only noted low BSO toxicity in primary fibroblasts. In this section, please also elucidate figures 2c and 2d.
2. Please explain in detail why the authors chose the patient with LHON, a genetic mitochondrial illness that involves a mutation in complex 1, and how this mutation affects the other aspects of the study in result 2.2.
3. In the result 2.3, please elucidate figure 3d and 3e in this section.
4. Figure 4e's expression is ambiguous, please consider a clearer manifestation.
5. Rather than using references, please describe the chemicals in the method 4.1 in considerable detail.
6. This paper takes different incubation times for erastin-treated MRC5-SV40 cells in different figures. Please show the standard of the time chosen, and the incubation times should be consistent. If not, explain the rationale in the discussion.
7. Please think about including the conclusion of this article and the directions suggestions for future research should go on this topic in a CONCLUSION section.
Author Response
- ”There are more recent articles about BSO, please either update this reference or explain why the authors' findings in result 2.2 are incongruent with reference 20, which only noted low BSO toxicity in primary fibroblasts. In this section, please also elucidate figures 2c and 2d”.
The new references concerning BSO-induced toxicity in fibroblasts from patients with other mitochondrial diseases were added to the revised version. Hypersensitivity to BSO has been described in fibroblasts from patients with Leigh's syndrome caused by a mutation in SURF1 (a gene encoding a mitochondrial chaperone involved in cytochrome oxidase assembly) or two mutations in the ND3 and NDUFA1 subunits of complex I, as well as in fibroblasts of patients with MELAS (myopathy, encephalopathy, lactic acidosis and stroke-like episodes) caused by mutations in two mitochondrial tRNAs.
The legend to Fig. 2C, D was improved.
2 “Please explain in detail why the authors chose the patient with LHON, a genetic mitochondrial illness that involves a mutation in complex 1, and how this mutation affects the other aspects of the study in result 2.2”
We have chosen these cells since (1) we have earlier reported higher level of oxidative stress in untreated cells, (2) in fibroblasts from patients with other mitochondrial diseases associated with mutations in Complex I ( Leigh's syndrome, MELAS) high sensitivity to BSO has been described.
- 3. In the result 2.3, please elucidate figure 3d and 3e in this section.
The legends to Fig. 3D, E were improved.
- Figure 4e's expression is ambiguous, please consider a clearer manifestation.
On Fig. 4E we have added data on the induction of hydrogen peroxide formation by succinate in the presence of antimycine and improved the legend.
5.”Rather than using references, please describe the chemicals in the method 4.1 in considerable detail”.
Done
6.“This paper takes different incubation times for erastin-treated MRC5-SV40 cells in different figures. Please show the standard of the time chosen, and the incubation times should be consistent. If not, explain the rationale in the discussion”.
In the experiments concerning cell viability we have chosen 24h time point since it was enough to reach significant death rate. To study lipid peroxidation we have analyzed only viable cells co the time of incubation was shorter. This is explained in the revised version.
- Please think about including the conclusion of this article and the directions suggestions for future research should go on this topic in a CONCLUSION section.
Done
Round 2
Reviewer 2 Report
The paper can be accepted without any further changes.